# Can Hydrogen Water Enhance Oxygen Saturation in Patients with Chronic Lung Disease? A Non-Randomized, Observational Pilot Study

**DOI:** 10.3390/diseases11040127

**Published:** 2023-09-26

**Authors:** Ram B. Singh, Alex Tarnava, Ghizal Fatima, Jan Fedacko, Viliam Mojto, Tyler W. LeBaron

**Affiliations:** 1Department of Medicine, Halberg Hospital and Research Institute, Moradabad 244001, India; 2Natural Wellness Now Health Products Inc., Maple Ridge, BC V4R 2S6, Canada; 3Department of Biotechnology, Era’s Lucknow Medical College and Hospital, Lucknow 226003, India; 4Centre of Clinical and Preclinical Research-MEDIPARK, Pavol Jozef Safarik University, 040 11 Kosice, Slovakia; 5Third Department of Internal Medicine, Faculty of Medicine, Comenius University, 813 72 Bratislava, Slovakia; 6Molecular Hydrogen Institute, Cedar City, UT 84720, USA; 7Department of Kinesiology and Outdoor Recreation, Southern Utah University, Cedar, UT 84720, USA

**Keywords:** antioxidant, COVD-19, oxidative stress, inflammation, hydrogen-rich water, hypoxia, COPD

## Abstract

Background: Recently, chronic lung diseases have been found to be associated with marked inflammation and oxidative stress, which leads to fibrosis in the lungs and chronic respiratory failure. This study aims to determine if hydrogen-rich water (HRW) can enhance oxygen saturation among patients with chronic lung diseases. Methods: Ten patients with chronic lung diseases due to COPD (*n* = 7), bronchial asthma (*n* = 2), and tuberculosis of the lung (*n* = 1) with oxygen saturation of 90–95% were provided high-concentration (>5 mM) HRW using H_2_-producing tablets for 4 weeks. Oxygen saturation was measured via oximeter and blood pressure via digital automatic BP recorder. Results: HRW administration was associated with a significant increase in oxygen saturation (SpO_2_) and decrease in TBARS, MDA, and diene conjugates, with an increase in vitamin E and nitrite levels, compared to baseline levels. Physical training carried out after HRW therapy appeared to increase exercise tolerance and decrease hypoxia, as well as delay the need for oxygen therapy. Conclusion: Treatment with HRW in patients with hypoxia from chronic lung diseases may decrease oxidative stress and improve oxygen saturation in some patients. HRW therapy may also provide increased exercise tolerance in patients with chronic hypoxia, but further research is needed.

## 1. Introduction

Chronic respiratory diseases (CRD) affect the airways and other structures of the lungs. Some of the most common are chronic obstructive pulmonary disease (COPD), bronchial asthma, occupational lung diseases, tuberculosis, cancer, and pulmonary hypertension [1]. All the chronic lung diseases such as bronchial asthma and COPD, as well as post-COVID-19 pulmonary fibrosis, may be associated with lung damage along with fibrosis and emphysema. These are major causes of chronic respiratory failure and subsequent hypoxia. It is well known that the conditions of severe oxygen deprivation have lethal consequences. The aim of the WHO CRD program is to support various countries in their efforts to reduce the toll of morbidity, disability, and premature mortality related to CRDs, specifically asthma and COPD [1]. One of the primary causes of COPD is smoking due to the high levels of toxic ingredients that are inhaled and ingested into the body. However, many who do not smoke can still be at risk due to environmental pollutions, which have also been shown to be a major cause of COPD [2]. There is evidence that both smoking, and pollution can cause a deficiency of antioxidants resulting in oxidative stress in the tissues leading to inflammation and lung damage [3,4].

Since there is no cure for these diseases, new therapies that reduce hypoxia are urgently needed. Experimental and clinical studies published in the last 15 years have indicated that molecular hydrogen (H_2_ gas) has therapeutic effects in diverse conditions, including metabolic syndrome [5]. It seems that any injury induced by a disease such as cancer or radiation in any of the body systems such as the respiratory system, nervous system, reproductive system, etc., are all areas which may benefit from hydrogen therapy [5,6,7]. It is hypothesized that molecular hydrogen therapy may decrease inflammation, oxidative stress, and reduce hypoxia [6,7]. In 2007, molecular hydrogen was demonstrated to act as a therapeutic antioxidant in preventive and therapeutic applications [8]. Researchers induced an ischemia reperfusion injury in a rat model of stroke by occluding the middle cerebral artery. The rats underwent hypoxia (artery occlusion) for 90 min and then reperfusion for 30 min. During the entire process, rats inhaled 2% H_2_ gas. Compared to control rats, the inhalation of molecular hydrogen significantly suppressed brain injury and reduced oxidative stress. The inhalation of molecular hydrogen was as effective as the drug, FK506, and more effective than edaravone, which are known ROS scavengers used in the treatment of cerebral infarction. Additionally, they demonstrated that molecular hydrogen dissolved in cell culture at physiologically relevant concentrations that correspond to the concentration that is achieved by inhalation (i.e., 25 µM to 400 µM) was able to reduce levels of hydroxyl radicals (^•^OH) and to a lesser extent, peroxynitrite (ONOO^−^). However, it did not react with physiologically important ROS including hydrogen peroxide (H_2_O_2_), superoxide (O_2_^−^), or nitric oxide (NO^•^) [8]. These results indicate that hydrogen could be therapeutic in conditions of oxidative stress, which has been confirmed by many subsequent studies in a variety of diseases and animal disease models [7].

In was demonstrated that administration of molecular hydrogen had a protective effect in a mouse model of cigarette-smoke-induced COPD [9]. Senescence marker protein 30 knockout mice were exposed to cigarette smoke for eight weeks. However, administration of water infused with molecular hydrogen significantly reduced the destructive index of the lungs, and significantly attenuated the cigarette smoke induced decrease in lung compliance. Mechanistically, it was reported that molecular hydrogen decreased markers of DNA oxidation such as phosphorylated histone H2AX and 8-hydroxy-2′-deoxyguanosine, and senescence markers such as cyclin-dependent kinase inhibitor 2A, cyclin-dependent kinase inhibitor 1, and β-galactosidase. These results indicate that molecular hydrogen may be a novel preventive and therapeutic approach for COPD [9].

In a pilot study, hydrogen gas inhalation ameliorates airway inflammation in bronchial asthma and COPD patients [7]. A case report published earlier suggests that supplementation with hydrogen-rich water (HRW) increases oxygen saturation up to 45 min after its administration in a patient with COVID-19-like symptoms [10]. The subject was suffering from chronic hypoxia with oxygen saturation between 91% and 93% and had persistent fibrosis of the lower long. Despite being treated with 100 mg of Nintedanib twice daily for 45 days as well as ingesting 200 mg of CoQ10 and 5 mg of apixaban and rosuvastatin, there was still no improvement in oxygen saturation. The patient was then administered high-concentration hydrogen-infused water twice daily. Treatment with molecular hydrogen was associated with increased oxygen saturation, increased exercise tolerance, and improved quality of life [10]. However, there are still gaps in the knowledge on clinical efficacy of molecular hydrogen in various chronic lung diseases [7]. For example, whether HRW has a therapeutic role on human diseases with airflow limitation. This pilot observational study aims to investigate the role of molecular hydrogen in patients with chronic lung diseases, in particular COPD.

## 2. Subjects and Methods

Ten patients of Indian ethnicity (7 men and 3 women; age 63.1 ± 8.3) with chronic lung diseases due to COPD (*n* = 7), bronchial asthma (*n* = 2), and chronic pulmonary tuberculosis (*n* = 1) with oxygen saturation <95% were prospectively observed to look for any effects of molecular hydrogen on clinical parameters of chronic lung diseases. Permission for this investigation, reviewing records, and publication was obtained from the ethics committee of the Halberg Hospital and Research Institute, Moradabad, India (Protocol #15012022). Records of patients were obtained from the department of Medicine of this hospital. Since there is no curable treatment available for bronchial asthma, COPD, or lung fibrosis, all the patients were advised to take molecular hydrogen therapy for possible benefits. Informed consent was taken from each patient (*n* = 10) for administration of molecular hydrogen.

Inclusion criteria for patients to be selected into this prospective observational study was that they did not respond to conventional treatment for 7–10 days. That is, they still had an oxygen saturation below 94%. Treatment included antibiotics, antiallergic (chlorpheniramine or bilastine) and montelukast, bronchodilators (salbutamol, levolin), and budecort (Budesonide) nebulizers in standard doses. Clinical, radiological data, and past family history of diseases were recorded. Oxygen saturation was measured via oximeter and blood pressure via digital automatic BP recorder.

The criteria for the diagnosis of chronic respiratory failure were based on a history of episodes of cough, breathlessness, with or without low-grade fever, decline in pulmonary function, forced expiratory volume 1(FEV 1), <50–80%, in conjunction with oxygen saturation <95%, and radiological examination showing fibrosis of lung parenchyma. FEV1 indicates the amount of air that a person can force out of their lungs in 1 s via spirometer. Detailed pulmonary function tests were not available.

Clinical data including body mass, body mass index, and blood pressure were recorded (Table 1). Blood pressure measurements, specifically systolic and diastolic values corresponding to Phase V of the Korotkoff sounds, in the right arm of participants were recorded. These measurements were taken following a 5 min period of seated rest. A single standard mercury manometer was employed for this purpose, with all assessments conducted by a consistent physician across all subjects. Additionally, participants’ body weights were measured independently by a healthcare professional, with individuals wearing undergarments, and results were recorded to the nearest 0.5 kg. Waist and hip girth measurements were obtained from participants while they were in a standing position. Hypertension was diagnosed in accordance with established guidelines, wherein systolic blood pressure readings equal to or exceeding 140 mm Hg and diastolic blood pressure readings equal to or exceeding 90 mm Hg were considered indicative of hypertension, consistent with recommendations from relevant agencies.

We additionally monitored standard various biochemical data including fasting blood glucose, thiobarbituric acid reactive substances (TBARS), malondialdehyde (MDA), diene conjugates, vitamins E and C, and nitrite. These were measured via colorimetric methods using a UV–VIS Spectrophotometer (Electronics Corporation of India, Ltd., Hyderabad, India) as described previously [5]. These data were measured at baseline and again after 4 weeks of HRW administration.

### HRW Administration

HRW was prepared using hydrogen-producing tablets, which react with water to produce molecular hydrogen according to the following reaction (Mg + H_2_O → H_2_ + Mg(OH)_2_, as described previously [5,9]. In brief, a tablet was dissolved in 200–300 mL of water (3 tablets in morning and 2 in the evening). Participants were instructed to gulp water immediately upon tablet disintegration. The hydrogen-producing tablets were supplied by HRW Natural Health Products Inc., New Westminster, BC, Canada. Total hydrogen intake is estimated at >10 millimoles/day. Hydrogen concentration reaches a peak of ~8 mM in 250 mL of water, as per gas chromatography (SRI 8610C; Los Angeles CA, USA) results from H2 Analytics (Henderson, NV, USA).

Statistical Analysis.

The prevalence rates are given in percent and continuous variables as mean plus or minus 1 standard deviation (mean ± SD) and ordinal variables as percentages. The assessment of the significance of associations among diverse risk factors was conducted through regression analysis. This analysis involved the calculation of odds ratios and corresponding 95% confidence intervals, employing both univariate and multivariate approaches. The dependent variable for this analysis was the overall prevalence of coronary artery disease. Significance was established at a *p*-value of less than 0.05, utilizing a two-tailed *t*-test as the criterion for statistical significance.

## 3. Results

The clinical data including age, sex, blood pressure are presented in Table 1. The ages ranged between 53 to 67 years, with 8/10 of the participants being male. The frequency of COPD was 70% (*n* = 7), with additional participants having bronchial asthma (*n* = 2) and chronic pulmonary tuberculosis with fibrosis (*n* = 1). Half of the patients had hypertension (BP ≥ 140/90 mm Hg) and two patients had borderline diabetes mellitus (random blood glucose ≥ 126 mg/dL). All the patients (90.0%), except one, had a history of smoking, 10 to 15 cigarettes daily for the past 5–15 years.

Radiological examination revealed prominent broncho-vascular marks with hyperinflated lungs, which appeared larger than normal. One patient who had pulmonary tuberculosis, showed a white shadow of fibrosis in the lung.

Table 2 shows oxygen saturation in the individual patients after 4 weeks of HRW treatment for 5 min, 30 min, and 45 min. Of the 10 patients, 7 showed improved oxygen saturation following four weeks of hydrogen therapy. However, one patient with tuberculosis and fibrosis of the lungs and two with COPD showed no benefit.

Effects of HRW on a variety of plasma biomarkers are presented in Table 3. Nighttime blood glucose, thiobarbituric acid reactive substances (TBARS), Malondialdehyde (MDA), and diene conjugates showed significant decline after hydrogen therapy. However, vitamin E, vitamin C, and nitrite levels showed a significant increase after 4 week of hydrogen administration.

## 4. Discussion

COPD and other lung diseases have hallmark signatures of increased oxidative stress and a dysregulation of antioxidant and oxidant balance [1,2]. Improving the antioxidant status and preventing oxidative stress may help reduce COPD symptoms, the progression of the disease, and allow the body to recover [2,3]. Molecular hydrogen represents a novel antioxidant that can help prevent oxidative stress [5,8]. Moreover, due to the small size and the neutral and nonpolar nature of molecular hydrogen, it can easily penetrate the biomembranes of cells and mitochondria [8]. Importantly, H_2_ gas has no known toxic side effects even at very high concentrations/doses [5,6]. The biological safety of H_2_ has been demonstrated in numerous studies dating back to the 1940s where molecular hydrogen was used to prevent decompression sickness in deep-sea diving [6]. H_2_ is also naturally produced by the gut microbiome upon metabolizing non-digestible carbohydrates [6]. Thus, H_2_, is uniquely qualified as a natural and safe antioxidant to play an important role in combating a variety of diseases.

This study indicates that treatment with hydrogen-rich water (HRW) in patients with chronic lung disease suffering from hypoxia (O_2_ saturation < 95%) was beneficial in improving oxygen saturation in most of the patients (*n* = 7, 70.0%). Apart from the observed improvements in symptoms and oxygen saturation, parameters of oxidative stress such as thiobarbituric acid reactive substances (TBARS), malondialdehyde (MDA), and diene conjugates showed a significant decline. In accordance with these effects, the antioxidants, vitamins E and C, showed a significant increase following HRW therapy. These results indicate that administration of molecular hydrogen may help reduce oxidative stress. There may be several ways to explain these observations on oxidative biomarkers. The antioxidant activity of molecular hydrogen may have prevented the oxidation of vitamins A and C, which would result in their increased concentration in the plasma. Alternatively, molecular hydrogen may have improved antioxidant/redox cycling, leading to the improved availability of antioxidants to antagonize the oxidative stress [6]. It has been proposed that an increase in oxidative stress is associated with an increased consumption of available antioxidants in tissue, with a decline in endogenous antioxidants such as superoxide dismutase, catalase, and ceruloplasmin along with exogenous antioxidants (e.g., vitamin A, E, C and beta-carotene, coenzyme Q10, etc.) [3,4,5]. However, an increased intake of antioxidants may cause a rise in both endogenous and exogenous antioxidants because they are spared from increased consumption due to oxidative stress [3,4,5]. Since, molecular hydrogen is a potential antioxidant and anti-inflammatory agent [5,6,7,8], supplementation with HRW may have attenuated excessive oxidative stress, thereby preventing a decline in antioxidant status. A decrease in oxidative stress and inflammation in the lung tissue may be responsible for the improvement in symptoms as well as oxygen saturation [7,10].

In a recent preliminary investigation, a gas mixture containing 2.4% hydrogen was administered via inhalation for a duration of 45 min to a cohort comprising 10 patients with asthma and 10 patients with chronic obstructive pulmonary disease (COPD) [7]. Following a single session of hydrogen inhalation, a notable reduction in monocyte chemotactic protein 1 levels was observed in both the COPD group (564.70–451.51 pg/mL, *p* = 0.019) and the asthma group (386.39–332.76 pg/mL, *p* = 0.033). Notably, a decrease in interleukin (IL)-8 levels was specifically noted in the asthma group (5.25–4.49 pg/mL, *p* = 0.023). Furthermore, in the COPD group, the concentration of soluble cluster of differentiation-40 ligand in exhaled breath condensate (EBC) increased subsequent to hydrogen inhalation (1.07–1.16 pg/mL, *p* = 0.031). Conversely, both the COPD (0.80–0.64 pg/mL, *p* = 0.025) and asthma (0.06–0.05 pg/mL, *p* = 0.007) groups exhibited significantly lower levels of interleukin-4 (IL-4) and interleukin-6 (IL-6) in EBC following hydrogen inhalation. These findings collectively indicate that a single 45 min session of hydrogen inhalation has the potential to ameliorate the inflammatory status within the airways of patients diagnosed with asthma and COPD [7].

A 2.4% H_2_ inhalation corresponds to a predicted blood H_2_ concentration, based on Henry’s Law, of around 17 µM. However, our study used administration of hydrogen water, which is a different method of administration. Nevertheless, the concentration of molecular hydrogen in the blood is expected to reach 25 µM to over 50 µM based on ingesting approximately two millimoles of molecular hydrogen. This concentration has been shown in cell culture studies to significantly reduce toxic hydroxyl radicals as assessed via reductions in the fluorescence signal emitted by the oxidized form of 2-[6-(4′-hydroxy)phenoxy-3*H*-xanthen-3-on-9-yl] benzoate (HPF) [8]. However, despite this impressive result it has been difficult to conceive how molecular hydrogen could react with hydroxyl radicals to provide any biological benefit in animal and human studies due to the slow reaction rate constants compared to endogenous antioxidants that are in greater abundance [5,8]. This apparent discrepancy may be reconciled by a recent report which identifies Fe-porphyrin as a redox sensor and biocatalyst of molecular hydrogen [11]. It was found that Fe-porphyrin is rich cytochromes such as in the mitochondria, and the hemoglobin of red blood cells, and thus ubiquitously distributed throughout the body. A reaction was confirmed between hematin and molecular hydrogen via ^1^H NMR and FTIR analysis. In this way, Fe-porphyrin can act as biocatalyst to increase the reaction rate between molecular hydrogen and strong oxidants. Moreover, the authors reported that the Fe-porphyrin structure can induce the hydrogenation/reduction of CO_2_ by molecular hydrogen into small levels of carbon monoxide, which is known to have numerous therapeutic biological effects. Additionally, it was shown that the Fe-porphyrin in hemoglobin can catalytically induce the reduction of molecular oxygen to water. The findings of CO production and reduction of O_2_ by molecular hydrogen suggest that hydrogen therapy might decrease oxygen levels [11]. However, this hypothetical is at odds with our study and previous investigations [7,10] in which hydrogen therapy was associated with increased oxygen levels. This might be explained by a recent study demonstrating that in rats with chronic heart failure, inhalation of molecular hydrogen led to increased erythrocyte ATP production [12] [9a]. The study also found increased levels of 2,3 bisphosphoglycerate, which resulted in improved microcirculation and oxygen transport function of the blood [12].

It seems that the COPD encompasses chronic bronchitis, emphysema, and small airway obstruction due to environmental exposures, primarily cigarette smoking. It is possible that genetic factors interact with smoking among patients with family history of bronchial asthma and other lung diseases in developing COPD for many patients in the population. It is also possible that host factors such as Western diet, sleep disorders, and alcoholism also interact with the environmental pollution to increase the propensity to develop chronic respiratory diseases [1,2]. Major pathogenic factors that contribute to developing respiratory disease include infection and inflammation, protease and antiprotease imbalance, and oxidative stress, which overwhelms the antioxidant defenses. Moreover, reactive oxygen species play a pivotal role in the incidence of acute exacerbations of diseases [1,2,3,4]. The major feature of COPD and other united airway diseases is characterized by an abnormal response to infection or allergen-induced injury, causing chronic inflammation, and subsequent activation of macrophages, eosinophils, neutrophils, T lymphocytes, and fibroblasts in the lung. Tobacco smoking or pollution produces a shift in the normal balance between oxidants and antioxidants in the body tissues, impacting oxidative stress both in the lungs and body systems [1,2,3,4]. In cigarette smoke, oxidants can directly injure cells and tissues, inactivate defense mechanisms, and initiate inflammation, which further increases oxidative stress [3,4]. The complexity of COPD and bronchial asthma necessitate a multi-target therapeutic approach which can influence inflammation in the lung tissues. It is likely that antioxidant supplementation such as hydrogen therapy, and dietary antioxidants may have a place in future combination therapies for a variety of chronic respiratory diseases [6,7,8,9,10].

In this study we also found that serum levels of nitrite showed an increase, indicating an increase in nitric oxide (NO^•^) production, which has inflammatory and anti-inflammatory effects [13]. In a larger study, we also found significant increase in serum nitrite after treatment with hydrogen from 0.63 ± 0.06 to 0.68 ± 0.06 μM [14]. It is possible that decline in nitrite can also occur during nitrosative or oxidative stress, resulting in a decrease in nitrite in the blood. However, increased levels of NO^•^ can also contribute to nitrosative stress due to the spontaneous reaction with superoxide to form toxic peroxynitrite [13]. These metabolic alterations decrease the ½ life of nitric oxide, resulting in a decline in the NO^•^ benefits and in cellular damage. Diverging from traditional antioxidants and anti-inflammatory agents, molecular hydrogen has exhibited distinctive antioxidant and anti-inflammatory properties. Its actions are characterized by a selective reduction in excessive inflammation and the mitigation of toxic oxidants, all while preserving crucial signaling reactive oxygen species (ROS) such as NO^•^. It has been suggested that hydrogen therapy may regulate NO^•^ production by increasing its circulating ½ life, potentiate the bioactivity of NO^•^, and act may even act as a NO^•^ mimetic by increasing cGMP levels [13]. Moreover, hydrogen can also prevent peroxynitrite formation and reduce the adverse effects from NO^•^ metabolism, such as lower nitrotyrosine levels [13]. HRW therapy may also provide increased exercise tolerance in patients with chronic hypoxia [10]. Recently, molecular hydrogen has been suggested for the treatment of COVID-19 [13,15,16,17], as well as COPD [18] due to its potential antioxidant and anti-inflammatory effects.

In a randomized, double-blinded, placebo-controlled trial involving 10 centers, patients (*n* = 54 in each group) with acute exacerbation of COPD (AECOPD) and a Breathlessness, Cough, and Sputum Scale (BCSS) score of at least 6 points were randomly assigned to receive a hydrogen/oxygen therapy or oxygen only [18]. Improvements in the BCSS score in the hydrogen/oxygen group was significantly greater compared to the oxygen only group (−5.3 vs. −2.4 point; difference: −2.75 [95% CI −3.27 to −2.22], indicating superiority. Other time points from day 2 through day 6 had similar findings. The Cough Assessment Test score exhibited a noteworthy reduction in the hydrogen/oxygen group compared to the control group (−11.00 vs. −6.00, *p* < 0.001). Conversely, alterations in pulmonary function, arterial blood gas levels, noninvasive oxygen saturation, and various other endpoints did not display significant differences between the two groups. However, similar to our findings, analysis of the per-protocol set did find significant group interactions with respect to the changes from baseline in SpO_2_ (*p* < 0.0001). Acute exacerbations were reported in 34 (63.0%) patients in the hydrogen/oxygen group and 42 (77.8%) in the oxygen group. This trial demonstrated that hydrogen/oxygen therapy is superior to oxygen therapy in patients with acute exacerbation of COPD, with an acceptable safety and tolerability profile [18].

COPD is one of the most common chronic illnesses in the world [1,2] which is associated with low-grade systemic inflammation, characterized by an increased production of pro-inflammatory cytokines, reactive oxygen species (ROS), and reactive nitrogen species [3,4]. Beyond these pro-inflammatory molecules, deficiencies in eicosanoids, reductions in endogenous antioxidants, anti-inflammatory cytokines, and specific long-chain polyunsaturated fatty acids (PUFA), along with their anti-inflammatory derivatives including lipoxins, resolvins, protectins, maresins, and nitrolipids, may also be present. The observed imbalance between pro-inflammatory and anti-inflammatory factors in these conditions suggests that therapeutic approaches aimed at suppressing unwarranted inflammation while promoting the synthesis or efficacy of anti-inflammatory bioactive lipids may offer valuable contributions to the prevention and management of COPDs and potentially other chronic diseases [14]. It is possible that hydrogen therapy may decrease inflammation by reducing free-radical-induced damage, which may decrease hypoxia and delay the need for oxygen therapy.

There is a limitation in the ability of current medical therapies to reverse severe respiratory failure in COPD patients. Acute exacerbation of COPD, causing persistent hypercapnia, may result in the need for early re-hospitalization with greater mortality [19]. It seems that oxygen therapy may induce hypercapnia, which is a common complication in patients with COPD [19]. There is evidence that noninvasive ventilation can improve outcomes in patients with COPD and acute respiratory failure [19]. Thus, an alternative therapy such as molecular hydrogen is likely to improve the effects of oxygen therapy. The pharmaceutical gaseous molecules, known as medical gases, including oxygen, nitric oxide, molecular hydrogen, and helium are emerging as novel and innovative therapeutic tools for COPD [17,20]. Interestingly, a controlled trial has shown that treatment with nitric oxide can decrease pulmonary hypertension in COPD patients [21]. Apart from this, in severe COPD, therapy with helium inhalation can be helpful in the reversal of airflow obstruction by reducing the resistance to flow in the airways and the work of breathing [20,22]. Helium gas is known to have a low density and molecular weight (MW). However, molecular hydrogen, which has a lower molecular weight, has been suggested to be a potential therapy for preventive and therapeutic applications against many diseases because in addition to its rapid diffusion, it has protective antioxidant, anti-inflammatory, anti-apoptotic effects [23,24]. Studies have shown that molecular hydrogen, as the lightest and smallest gas molecule, has a unique function as an antioxidant to improve lung function [7,24,25]. Therefore, it has been proposed that the inhalation of a hydrogen/oxygen mixture may be an alternative therapy for COPD [7,15]. The efficacy of hydrogen/oxygen therapy in patients with tracheal stenosis has already been demonstrated [26]. The combination of oxygen and hydrogen inhalation in patients with acute exacerbation of COPD may be useful, but the use of HRW may be more convenient, as indicated by this pilot study. However, despite the apparent favorable effects of HRW in our study, additional research is needed. For example, HRW therapy should be investigated without using other therapies. This could not be carried out in this study as it was more observational, and we could not risk the exacerbation of the condition. Furthermore, this was a small study, hence caution is warranted when interpreting these results. Finally, in addition to larger studies that are double-blinded and placebo-controlled, the clinical evidence for the efficacy of HRW therapy compared to H_2_ inhalation therapy in patients with COPD should also be investigated. This may prove important as some COPD patients may have fluid restrictions, which may pose difficulty in administering sufficient H_2_ through HRW, especially if the patient has different fluid intake preferences.

## 5. Conclusions

In conclusion, this study indicates that 4-week treatment with HRW can significantly decrease oxidative stress while also increasing oxygen saturation among patients with chronic lung diseases. However, larger studies using a double-blinded, placebo-controlled design are needed before making strong conclusions about the benefits and efficacy of HRW therapy. Additionally, studies comparing the effects of HRW and hydrogen inhalation with or without oxygen should be conducted.

## Figures and Tables

**Table 1 diseases-11-00127-t001:** Clinical data at baseline among all the patients with chronic lung disease.

Data	Measurement, Mean ± Standard Deviation
Sex, male, number (%)	8 (80%)
Mean age, years	63.1 ± 8.3
Mean body weight, kg	58.4 ± 4.4
Mean body mass index (BMI), Kg/M^2^	64 ± 4.4
Blood Pressure (systolic) mm Hg	138.1 ± 6.2
Blood Pressure (diastolic) mm Hg	88.3 ± 2.4

**Table 2 diseases-11-00127-t002:** O_2_ saturation in individual patients at baseline and after treatment with HRW.

Diagnosis	SpO_2_ Saturation				Results
	Baseline	5 min	30 min	45 min	Improved?
Chronic obstructive pulmonary disease (COPD)	90	95	94	94	Yes
Bronchial asthma	94	96	95	94	Yes
COPD	95	95	94	95	No
Tuberculosis lungs	92	92	92	91	No
COPD	94	95	95	94	Yes
Bronchial asthma	95	96	96	95	Yes
COPD	95	95	95	95	No
COPD	93	94	94	93	Yes
COPD	93	94	94	93	Yes
COPD	92	94	94	93	Yes
Total	93.3	94.6	94.3	93.7	7/10 improved

**Table 3 diseases-11-00127-t003:** Effect of hydrogen rich water on oxidative stress and antioxidants.

Biomarkers	Baseline	After 4 Weeks	*p*-Value
Fasting blood glucose, mg/dL	88.11 ± 4.8	85.49 ± 3.9	0.20
Night blood glucose, mg/dL	108.3 ± 3.9	103 ± 3.2	0.008
TBARS, µg/mL	2.6 ± 0.1	2.1 ± 0.2	0.0001
Malondialdehyde, µg/mL	3.2 ± 2.0	2.7 ± 0.2	0.0004
Diene conjugates, <25 OD	27.3 ± 0.7	25.9 ± 0.7	0.0005
Vitamin E, μM	21.9 ± 1.3	23.6 ± 0.5	0.002
Vitamin C, μM	19.9 ± 1.3	22.0 ± 0.7	0.005
Nitrite, μM	0.6 ± 0.02	0.68 ± 0.04	0.0002

*p* value was obtained via Student’s *t*-test by comparing baseline data with data after treatment. TBARS = Thiobarbituric acid reactive substances.

## Data Availability

Data are available on request to R.B.S.

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
