# Peer review of "Can Hydrogen Water Enhance Oxygen Saturation in Patients with Chronic Lung Disease? A Non-Randomized, Observational Pilot Study"

_diseases, 2023, doi:10.3390/diseases11040127_

Round 1
Reviewer 1 Report
The title of manuscript is good. English language has good quality. There are some explainations about different parts of the manuscript.
1. About section "2. Materials and Methods"
+ According to which criteria patients were diagnosed with COPD?
+ Please mention the findings of spirometry tests and also radiological findings in patients that have been undergone observation in present study
+ In the process of study, have patients delivered any other medication other than HRW?
2. In page 10, line 112-113
The aithors have mentioned that "they did not
respond to conventional treatment for 7-10 days"
What do you mean by "did not respond to conventional treatment"? Please explain in detail
3. Why you have measured body mass and body wieght in patients in your present study?
4. Please write about future insights and obstacles of your present work
5. Please write a separate section about serious liabilities and constraints of using HRW in COPD patients
6. You have observed 10 patients for your survey. Why you have not enroll more patients in your study? and also why you have not join female patients in your survey? Please explain
7. Please check and adjust the "Reference list" based on the regulations of reference list of journal. (Titles, doi, the name of journal and ... )
Author Response
The title of manuscript is good. English language has good quality. There are some explainations about different parts of the manuscript.
Response: Thank you for reviewing our article and providing useful feedback. We address your comments below.
1. About section "2. Materials and Methods" According to which criteria patients were diagnosed with COPD?
Response: We elaborated on the inclusion criteria in the methods section lines 129-134
Please mention the findings of spirometry tests and also radiological findings in patients that have been undergone observation in present study.
Response: Only forced expiratory volume was measured, without doing detailed pulmonary function tests. Radiological findings given under Results lines 176-179
In the process of study, have patients delivered any other medication other than HRW?
Response: Yes, we specify what other treatments were given in the methods section, lines 123-126
2. In page 10, line 112-113 The aithors have mentioned that "they did not respond to conventional treatment for 7-10 days" What do you mean by "did not respond to conventional treatment"? Please explain in detail.
Response: We included the types of treatments that were given in lines 129-134. Treatment included antibiotics, antiallergic (chlorpheniramine or bilastine) and montelukast, bronchodilators (salbutamol, levolin) and budecort (Budesonide) nebulizers in standard doses.
3. Why you have measured body mass and body wieght in patients in your present study?
Response: In our hospital setting this is a routine practice, as it is important for calculation of dosages of drugs.
4. Please write about future insights and obstacles of your present work.
Response: Yes, we added that the number of patients are small, hence larger randomized, controlled trials would be necessary to confirm the results of our pilot study and additional insights and obstacles for research in lines 381-391
5. Please write a separate section about serious liabilities and constraints of using HRW in COPD patients.
Response: We added the possible limitation/constraint of using HRW for COPD patients on fluid restriction and other considerations (lines 381-391). However, due to the high safety profile of molecular hydrogen, which section was also added (lines 208-215), there is little serious liability in using the H2 molecule as treatment
6. You have observed 10 patients for your survey. Why you have not enroll more patients in your study? and also why you have not join female patients in your survey? Please explain.
Response: Thank you for the observation. It is because this was merely a clinical observational of cases without any research grant which is necessary for a proper study. There were 3 females (see Line 111). We hope that this preliminary report can be used to procure additional funding in the future to make a larger study.
Reviewer 2 Report
In this paper, the authors investigated if hydrogen-rich water (HRW) can enhance oxygen saturation among ten patients with chronic lung diseases. Their results showed that after using HRW for 4 weeks, oxygen saturation (SpO2), vitamin E and nitrite levels were significantly increased, and TBARS, MDA, and diene conjugates were significantly decreased. This is an interesting study. However, some places of this manuscript need to be revised.
Here are some specific comments and suggestions:
Main Points:
The sample size in this clinical study (10 patients) is too small. The authors should consider recruiting a larger cohort of patients with chronic lung diseases to enhance the study's robustness and generalizability.
Specific Points:
1. This study is not directly related to COVID-19. Therefore, the descriptions about COVID-19 in the abstract should be removed.
2. In Table 1, please double-check the accuracy of the data for Mean Body Mass Index (BMI), specifically the value (164±4.4) to ensure its correctness.
3. Could you provide information about the levels of vitamin C after HRW treatment?
Author Response
Comments and Suggestions for Authors. In this paper, the authors investigated if hydrogen-rich water (HRW) can enhance oxygen saturation among ten patients with chronic lung diseases. Their results showed that after using HRW for 4 weeks, oxygen saturation (SpO2), vitamin E and nitrite levels were significantly increased, and TBARS, MDA, and diene conjugates were significantly decreased. This is an interesting study. However, some places of this manuscript need to be revised.
Response: Thank you for reviewing our article and providing useful feedback. We address your comments below.
Here are some specific comments and suggestions:
The sample size in this clinical study (10 patients) is too small. The authors should consider recruiting a larger cohort of patients with chronic lung diseases to enhance the study's robustness and generalizability.
Response: Yes, we agree. We added that this is a very small sample size, and more research is needed. Since this study is merely an observational report without any funding, we could not expand the study, but hope this can be done in the future. We added commentary to address the small size in lines 381-391
Specific Points:
- This study is not directly related to COVID-19. Therefore, the descriptions about COVID-19 in the abstract should be removed.
Response: Yes, we agree. It is removed.
- In Table 1, please double-check the accuracy of the data for Mean Body Mass Index (BMI), specifically the value (164±4.4) to ensure its correctness.
Thank you for noticing our typo. The correct value is 64±4.4
- Could you provide information about the levels of vitamin C after HRW treatment?
Response: Yes, we show this in Table 3, but since this was an observational and not a mechanistic study we did not do not provide detailed commentary as to why this was other than what we wrote in Lines 222-228
Round 2
Reviewer 1 Report
No more comment.